# Pesticide Use Practices in Root, Tuber, and Banana Crops by Smallholder Farmers in Rwanda and Burundi

**DOI:** 10.3390/ijerph16030400

**Published:** 2019-01-31

**Authors:** Joshua Sikhu Okonya, Athanasios Petsakos, Victor Suarez, Anastase Nduwayezu, Déo Kantungeko, Guy Blomme, James Peter Legg, Jürgen Kroschel

**Affiliations:** 1International Potato Center (CIP), P.O. Box 22274 Kampala, Uganda; 2International Potato Center (CIP), La Molina, Lima 12, Peru; t_petsakos@yahoo.gr (A.P.); V.SUAREZ@CGIAR.ORG (V.S.); 3Rwanda Agricultural Board (RAB), P.O. Box 73 Ruhengeri, Rwanda; anastasenduwa@yahoo.fr; 4International Institute of Tropical Agriculture (IITA), P.O. Box 1894 Bujumbura, Burundi; D.Kantungeko@cgiar.org; 5Bioversity International, c/o ILRI, P.O. Box 5689 Addis Ababa, Ethiopia; g.blomme@cgiar.org; 6International Institute of Tropical Agriculture (IITA), c/o AVRDC—The World Vegetable Center, P.O. Box 10 Duluti, Arusha, Tanzania; J.LEGG@CGIAR.ORG; 7International Potato Center (CIP), NASC Complex, DPS Marg, Pusa Campus, New Delhi 110012, India; J.KROSCHEL@CGIAR.ORG

**Keywords:** fungicides, insecticides, occupational health, personal protective equipment, poisoning, safety measures, training, integrated pest management

## Abstract

Misuse and poor handling of chemical pesticides in agriculture is hazardous to the health of farmers, consumers, and to the environment. We studied the pest and disease management practices and the type of pesticides used in four root, tuber, and banana (RTB) crops in Rwanda and Burundi through in-depth interviews with a total of 811 smallholder farmers. No chemical pesticides were used in banana in either Rwanda and Burundi, whereas the use of insecticides and fungicides in potato was quite frequent. Nearly all insecticides and about one third of the fungicides used are moderately hazardous. Personal protective equipment was used by less than a half of the interviewed farmers in both countries. Reported cases of death due to self- or accidental-poisoning among humans and domestic animals in the previous 12 months were substantial in both countries. Training of farmers and agrochemical retailers in safe use of pesticide and handling and, use of integrated pest management approaches to reduce pest and disease damage is recommended.

## 1. Introduction

Roots, tubers, and bananas (RTB) are important cash and food security crops in many countries in sub-Saharan Africa (SSA). Banana (*Musa* spp.), cassava (*Manihot esculenta* Crantz), potato (*Solanum tuberosum* L.), and sweetpotato (*Ipomoea batatas* L. Lam) are highly important staple crops in the livelihoods of smallholder farmers in the Great Lakes region of Central Africa. Consumption of orange-fleshed sweetpotato, yellow cassava, east African cooking banana and table potato in combination with iron-rich beans (*Phaseolus vulgaris* L.) help to reduce malnutrition but also increases household cash income through the sale of surplus food [1,2]. Average per capita consumption of sweetpotato in Rwanda (89 kg/pers/yr) is almost six times higher than the world average of 14 kg/pers/yr. Banana consumption in Rwanda (144 kg/pers/yr) is also the second highest in the world. In addition, Rwanda is ranked fifth in the world in consumption of cassava [3]. Potato consumption in Rwanda was estimated at 125 kg/pers/yr [4].

Despite the importance of RTB crops in the livelihoods of smallholder farmers, their production in Rwanda and Burundi is limited by numerous pests and diseases [3], which may cause yield losses of nearly 100% [5,6,7]. Most important are late blight caused by *Phytophthora infestans* (Mont.) de Bary, bacterial wilt caused by *Ralstonia solanacearum* Smith, aphids (*Aphis gossypii* Glover, *Aphis fabae* Scopoli, *Macrosiphum euphorbiae* Thomas, and *Myzus persicae* Sulzer) and potato tuber moth (*Phthorimaea operculella* (Zeller)) in potato; cassava mosaic disease (CMD), cassava brown streak disease (CBSD) and whitefly (*Bemisia tabaci* (Gennadius)) in cassava; Xanthomonas wilt of banana (BXW) caused by *Xanthomonas campestris* pv. *musacearum* (Yirgou & Bradbury 1968) Dye 1978 and Fusarium wilt caused by *Fusarium oxysporum* f. sp. *cubense* (E.F.Sm.) W.C. Snyder and H.N. Hansen in banana; sweetpotato virus disease (SPVD) and the African sweetpotato weevils (*Cylas puncticollis* Boheman and *C*. *brunneus* Olivier) in sweetpotato.

In response to high pest and disease pressure, farmers use several control measures to reduce yield- and post-harvest-losses including the application of pesticides. Unlike in non-commercial food crops, which command low prices in the local market, in cash crops in SSA like potato, coffee (*Coffea* spp.), cotton (*Gossypium hirsutum* L.), tomato (*Solanum lycopersicum* L.), eggplant (*Solanum melongena* L.), beans and a number of horticultural crops, farmers frequently use pesticides to control pests and diseases [8,9]. The proportion of large scale farmers (i.e. growing crops on 10 ha of land or more) using pesticides in Rwanda is increasing and was estimated at 46.7% of 195 in 2015 [10]. There is a consensus among farmers in Rwanda and Burundi that the frequency of pesticide application per cropping system has increased in recent decades due to increased prevalence of pests and diseases. Recent studies in both countries reported that more than half of farmers used insecticides in beans and tomato [11,12]. Pesticide use frequency on tomato was up to twice per week in Burundi with most farmers applying fungicides (Mancozeb 80%) directly on tomato fruits [12]. However, information on pesticide use practices in RTB crops in Rwanda and Burundi is not available, and there are few studies on the use of personal protective equipment (PPE), exposure symptoms, handling, and pesticide misuse.

Inspection of pesticide retail outlets or banning the importation of certain pesticides requires strict enforcement of pesticide legislation which needs an expensive monitoring process. This rarely receives enough national funding in many developing countries. Using hazardous pesticides involves several risks and requires knowledge of health and safety measures needed during pesticide application [13]. The risk of pesticide poisoning is also high when using leaking knapsack sprayers, purchasing pesticides in unlabeled containers, storing pesticides close to/with foodstuffs/food items or in the reach of children, and when PPE are used improperly. Due to the high risk of pesticide poisoning, adequate training of farmers and the use of PPE while handling pesticides is key. Socio-economic factors that may increase the risk of pesticide poisoning include: low income, incomplete formal education, poor knowledge of the negative effects of pesticide use, an inability to read and understand pesticide labels, as well as the reluctance of some farmers to use PPE. Poor PPE usage can result from a relaxed attitude to risk, the lack of adequate training in pesticide use as well as the gender of person spraying. 

Due to the negative effects of pesticides on human health [14,15,16], coupled with the need to introduce environmentally sustainable intervention measures, such as Integrated Pest Management (IPM), this article investigates and compares the use of pesticides among smallholder RTB farmers in Rwanda and Burundi using descriptive statistics and regression analysis. It attempts to fill the aforementioned information gap about pesticide use in these two countries and, to establish a baseline of current practices in RTB farming. Based on these findings, the study also identifies potential interventions to improve the efficiency of pesticide use and reduce the risk to farmers, consumers and the environment. This study is part of a larger project whose goal is to mitigate the likelihood of introduction, emergence, and spread of RTB pests and pathogens due to increased globalization of trade, human movement, farming practices, and climate change [17].

It was anticipated that results from this study would then feed into efforts to raise awareness creation of the need to enforce pesticide legislation and alternative control methods such as IPM. Findings of this study can be of significant value to several stakeholders in the pesticides value chain, including policy makers, public health professionals, vector control programs, agricultural extension workers, sellers of pesticides, and smallholder farmers.

Improper use of pesticides in the Kivu region has been linked to high levels of residues by persistent organic pollutants such as dichlorodiphenyltrichloroethane (DDT) in tilapia fish from Lake Tanganyika [18]. It has also been associated with severe reductions in populations of beneficial insects such as pollinator bees and parasitoids [19], contamination of surface water in Lake Kivu with malathion, metalaxyl and carbendazim [20], human diseases, and suicide [14,21]. Pesticide residues have been reported in breast milk, crops, water and body fluids in Australia, Ghana and Tanzania [22,23,24], fruits, and vegetables in Ghana [25].

Despite the harmful effects that can result from pesticide use, regulations on safe pesticide use are lacking in Rwanda and Burundi. This leaves decisions on pesticide storage, sale, packaging, labelling, transportation and handling to many untrained commercial sellers who may only be interested in maximizing profit, and who are not able to provide adequate information to pesticide users. However, even in other countries where laws exist, their enforcement at farm and retail level remains a challenge.

Information gained in this study can be used by public health professionals, policy makers, agricultural extension officers and research scientists when designing intervention programs on IPM including the safe use and handling of pesticides.

## 2. Materials and Methods

### 2.1. Survey Area and Tool 

The data were collected as part of a household survey carried out in Rwanda and Burundi using a questionnaire. The questionnaire was designed to capture the following: the type of personal protective equipment (PPE) used by farmers to avoid contact with pesticides during application, the signs and symptoms experienced during or after handling pesticides, knowledge of pesticides (doses, labels, toxicity), frequency of application, negative effects resulting from pesticide use (see part F on page 7–10 and part I on pages 13–14 of the questionnaire in Appendix A). Fifteen households per village were randomly selected for individual in-depth interviews and participation was entirely voluntary. The person in-charge of production of any of the four RTB crops within the household was the primary study subject. When a village didn’t have the 15 farming households, other households were selected from a neighboring village.

Individual face-to-face farmer interviews were conducted in two project action sites of Rwanda (Ruhengeri watershed comprising the districts of Musanze, Burera, Ngororero, Gakenke and Nyabihu) and Burundi (Rusizi watershed near Lake Tanganyinka comprising the provinces of Bubanza, Bujumbura Rural, Cibitoke and Muramvya) (Figure 1). A total of 811 households which had grown at least one of the four RTB crops (banana, cassava, potato, and sweetpotato) in the previous cropping season were interviewed. In collaboration with the Rwanda Agricultural Board (RAB) and Institute des Sciences Agronomique du Burundi (ISABU), the questionnaire was translated into Kinyarwanda for Rwanda and into French for Burundi. Administering the questionnaire in Burundi was in French and Kirundi while in Rwanda it was in Kinyarwanda. Enumerators were trained for two days and the questionnaire was pre-tested in districts outside of the survey area.

The household survey specifically aimed to assess farm household demographics and the existing pest and disease control methods with special emphasis on the use of pesticides, and their toxicity and application frequency. It also aimed to evaluate the protective measures used by farmers to reduce exposure to pesticides, to document the cases of acute poisoning experienced by farmers while handling pesticides and to determine the level of knowledge about pesticide handling and the degree to which PPE were used.

### 2.2. Ethical Statement 

Oral informed consent was sought from study participants after explaining the objectives of the study. Participation was therefore voluntary, and farmers were assured that the collected information will be treated fully confidentially. The farmers were also free to answer or decline any question or to withdraw from further participation in this interview at any time. It was also explained that declining or withdrawing from the interview would not have any negative consequence to the farmer or any household member and would not prevent him/her from benefitting from the results of the survey. An equivalent of the labor cost for one day was paid to the farmer after the interview as compensation for lost time.

### 2.3. Statistical Analysis

ANOVA and chi-square tests were used to analyze the survey data for descriptive parameters [26]. Further, we applied regression models to analyze relationships of different variables [27]. The estimation of how different variables like income and education determine the number of PPE used by farmers, constitutes a count data problem which is typically answered with a Poisson regression model. This model assumes that the dependent variable Y (event count) is a function of a vector of covariates x, and it is randomly drawn from a Poisson distribution:(1)$$P\left(Y=y|x\right)=\frac{e^{-\lambda }\lambda^{-y}}{y!}.$$

Parameter λ is the average event count (in our case the number of PPE in each farm household):(2)$$\lambda =E\left[y|x\right]=\exp\left(x^{⏉}\beta \right)$$
and β is the vector of the coefficients to be estimated. Since the conditional mean of Y is exponential, each element in β can be interpreted as a semi-elasticity, or a change in the logarithm of E[Y] for a unit change in the respective covariate, *ceteris paribus* [28]. 

One restrictive assumption of the Poisson model is its “equidispersion” property, in other words, the expected value of the count variable must equal its variance λ=E[Y]=VAR[Y]. This condition may not always hold in practice because the data often show more dispersion than what can be explained by the model. This over-dispersion problem can be addressed with a Negative Binomial (NB) regression model which derives from the Poisson model, under the additional assumption that λ is a random variable defined as $\lambda =\exp\left(x^{⏉}\beta \right)\nu$. The difference compared to equation (2) is the Γ-distributed dispersion term ν with E[ν]=1 and VAR[ν]=θ. With the addition of this stochastic dispersion term, the count variable Y now follows a negative binomial distribution Y~NB(λ,θ) for which the first two distribution moments are no longer equal, since E[Y]=λ (as in the Poisson) and $VAR\left[Y\right]=\lambda +\lambda^{2}/\theta$. Moreover, θ is a parameter that needs to be estimated. For a detailed analysis of the Poisson and NB models, the interested reader is referred to Cameron and Trivedi [28].

A second problem often encountered in both the Poisson and the NB models is their inability to correctly predict the zero counts that are observed in the sample. This shortcoming is usually the result of an excessive number of zeros in the count variable and requires a zero-inflated variant of the selected model [27]. Zero-inflated models consist of two distinct processes, one which produces the excess zero counts because of some specific data structure (structural zeros) and a second one which produces zero counts as a result of the underlying probability distribution (sampling zeros). Each data generating process is modelled with its own set of covariates (i.e. the two processes may be modeled using a different set of explanatory variables). For example, if structural zeros occur with probability π, then the probability mass function of the zero-inflated negative binomial model (ZINB) can be written as: (3)$$\{\begin{array}{l} P\left(Y=0\right)\;=\;\pi +\left(1-\pi \right){\left(\frac{\theta }{\theta +\lambda }\right)}^{\theta }\\ P\left(Y=y\right)\;=\;\left(1-\pi \right)\frac{\Gamma \left(y+\theta \right)}{\Gamma \left(\theta \right)\Gamma \left(y+1\right)}{\left(\frac{\theta }{\theta +\lambda }\right)}^{\theta }{\left(\frac{\lambda }{\theta +\lambda }\right)}^{y}\; \end{array}$$

The recommended procedure for selecting an appropriate count model is to first test for over-dispersion using a likelihood-ratio (LR) test and examine if θ is significantly different from zero [29]. If the zero hypothesis for θ is rejected, the NB model is more appropriate for describing the data generation process. Following the selection of the underlying distribution of the count variable, a Vuong test [30] was performed for examining zero-inflation.

## 3. Results

### 3.1. Pest and Disease Management Practices

Potato: In Burundi, most potato farmers (79.8%) used several cultural methods to control pests and diseases (Figure 2). Use of fungicides (55%) was significantly higher compared to insecticides (11.2%). Fungicides were applied to control late blight while insecticides were mainly sprayed to control leafminer flies (*Liriomyza* spp.) and potato aphids. A considerable number of interviewed farmers couldn’t specify the target insect pest (84.6%) or disease (36.6%), respectively. Herbicides were not mentioned by farmers as being used for weed control in potato. Combining different cultural practices was also an important farming practice in Rwanda (79.7%) to reduce pests and diseases. The use of insecticides was significantly higher in Rwanda compared to Burundi (41.2% vs. 11.2% of farmers, respectively). There was a similar pattern for fungicides (75.3% vs. 55%).

Sweetpotato: Cultural practices were widely used by farmers in Burundi (65.7%) to control sweetpotato pests and diseases (Figure 3). Few farmers (10%) did not control pests and diseases with only 14% applying insecticides to control the sweetpotato butterfly (*Acraea acerata* Hew.). Farmers’ used neither fungicides nor herbicides in both countries. The proportion of sweetpotato farmers using at least one cultural method was higher in Rwanda (83.1%) than Burundi (65.7%). 

Banana: Out of 244 and 209 banana farmers interviewed in Rwanda and Burundi respectively, none of them used any pesticide. However, 85.6% of banana farmers in Burundi and 75.4% in Rwanda used at least one cultural control method. 

Cassava: A few cassava farmers (2.3% in Burundi and 4% in Rwanda) applied chemical insecticides to control the cassava mealybug (*Phenacoccus manihoti* Matile-Ferrero), and cassava whiteflies (Figure 4). Use of at least one cultural control method in cassava was common in Rwanda (95.9%), as it was in Burundi (97.6%). 

### 3.2. Pesticides Used

#### 3.2.1. Active Ingredients and Toxicity Classes

The ten insecticides used by interviewed farmers in Rwanda were based on four active ingredients (chlorpyriphos, cypermethrin, profenofos, and dimethoate) either individually formulated or in combination and belong to the WHO Class II (moderately hazardous) (Table 1) [31]. Malathion dust, a slightly hazardous insecticide (WHO Class III), was used for protecting seed potato from damage by the potato tuber moth during storage. Using own farm saved potato seed from the previous harvestwas a common practice in both countries.

The eight types of fungicides reported in the survey in Rwanda were used exclusively for late blight control in potato, and consisted of the following active ingredients: mancozeb, metalaxyl, and benomyl. Metalaxyl is moderately hazardous (WHO Class II) while mancozeb and benomyl are both unlikely to present acute hazard in normal use (WHO Class U). Only two fungicides (Dithane and Ridomil) and two insecticides (Dursban and Rocket) were used in potato in Burundi. 

Most farmers didn’t know the trade names or active ingredients of the commonly used pesticides. Except for Ridomil, the rest of powder-based fungicides used in potato were locally referred to as Dithane, which was mostly sold in unlabeled transparent 0.5 kg plastic bags; in many cases it wasn’t possible to verify the true active ingredient. Insecticides were referred to as “simakombi” by farmers who didn’t know the exact trade name.

#### 3.2.2. Pesticides Application Frequency

In both Rwanda and Burundi, pesticides were applied with a hand-held mechanical knapsack sprayer. In Burundi, insecticides were always mixed with fungicides for applications in potato as a preventive measure. During a cropping season (3–4 months for potato, 3–12 month for sweetpotato and 6–12 months for cassava), the number of pesticide applications was highest among farmers of potato, for both fungicides and insecticides (10.2 ± 2.1) and lowest for insecticides (2.6 ± 0.2) among sweetpotato farmers in Burundi (Figure 5). In Rwanda, the number of fungicide and insecticide applications per season in potato were on average 6.7 and 5.0, respectively. 

#### 3.2.3. Use of Personal Protective Equipment (PPE) during Application

Less than a half of interviewed farmers in both Rwanda and Burundi used PPE (Figure 6). More than half of the farmers bathe after spraying in the two countries. No farmer wore eye goggles in Rwanda and only 4% reported the use of eye goggles in Burundi. Less than 10% of the farmers in both Rwanda and Burundi reported wearing hand gloves and waterproof jackets. Instead of face and nose masks, handkerchiefs were often used even though they give inadequate protection. Reasons for not using PPE were high cost (100% of farmers in Burundi, and 31% of farmers in Rwanda), unavailability at the local market (31% of the farmers in Rwanda), and no awareness about the use of appropriate PPE (35%); 18% of the farmers didn’t see any need to use PPE.

#### 3.2.4. Regression Model Results on PPE Use

The dependent variable (number of PPE—denoted as PPE in the model) was constructed as the aggregate sum of different protective gears used in the field. For instance, if a farmer used gloves and a mask, then PPE = 2, whereas if only a mask was used then PPE = 1. The underlying assumption is that the number of PPE is correlated with the level of protection during pesticide application. It is important to note that the survey did not distinguish between PPE ownership and actual usage. For what follows, we assume that ownership of PPE also implies use in the fields.

The variables examined for explaining the number of PPE, along with some descriptive statistics are given in Table 2. Among these variables, seven are continuous, five are binary and two are categorical. Variables FAM_EFF and OWN_EFF, i.e., the appearance of health issues from pesticide use by a family member or by the head of the household, did not distinguish between symptoms but they were categorized as “yes” if there was at least one symptom reported, and “no” otherwise.

The high standard deviation of the count variable in Table 2 is an indication of overdispersion. Using STATA (Release 14, StataCorp LP, College Station, Texas, USA), an LR test rejected the hypothesis that parameter θ is zero (*p*-value 0.0006) while the Vuong test suggested an excessive number of zeros in the data (*p*-value 0.0494), finally leading to the selection of a ZINB model. The country of origin (CTY) was chosen as the covariate for accounting for the excessive zeros in PPE use since exploratory analysis revealed that most farmers in Burundi do not use any type of PPE. The model was implemented in STATA and we used the robust option for estimating the regression coefficients. The results of the ZINB model are summarized in Table 3.

The estimated ZINB model was statistically significant (*p*-value of Wald chi-square < 0.05) and CTY adequately explained the excessive zeros in the count variable. Specifically, the logarithm of odds of having excess zeros for farmers in Burundi was 16 times greater compared to Rwanda. Despite causing zero inflation, CTY was also a statistically significant explanatory variable for the number of PPE as the logarithm of the PPE count for Burundi farmers was 0.73 higher that the PPE count in Rwanda. This result implies that Burundi farmers were less likely to use PPE, yet those who did, used a higher number of PPE compared to their Rwandan counterparts.

The number of PPE used was also affected by the size of the potato field (POT_FLD) and the frequency of the pesticide applications (APP_FRQ). Additionally, the logarithm of the PPE count was higher for those farmers who had personally experienced some symptom of poisoning (OWN_EFF). Interestingly, the model did not find any statistically significant impact of education, age, management training or amount of income on the number of PPE. On the contrary, the participation in farmers’ groups had a negative effect on the number of PPE used. This result could be attributed to sharing of PPE among many group members which in turn can reduce ownership.

#### 3.2.5. Symptoms after Pesticide Applications and Reported Consequences of Pesticide Poisoning 

The most commonly reported symptoms experienced by farmers in Burundi after pesticide applications were skin itching (60%), teary eyes (57%), burning eyes (53%), and reddened eyes (51%) (Figure 7). More farmers in Burundi had experienced some form of ill health symptoms than their counterparts in Rwanda.

In Rwanda, the five most commonly reported symptoms were runny nose (33%), headache (28%), coughing (25%), nausea (23%), and skin itching (21%). Less common symptoms of pesticide poisoning in Rwanda were stomach ache (2%), heavy sweating (4%) and perceived death of domestic animals after consumption of pesticide treated plants (4%). Perceived death cases among the interviewed farmers due to self-(suicidal) or accidental-poisoning for humans and domestic animals in the previous 12 months were substantial in both countries, i.e. 14 people and 10 animals in Rwanda and 11 people and 13 animals in Burundi.

#### 3.2.6. Various Pesticide Parameters

Retailers of agrochemicals were the main source for purchasing pesticides in Burundi (43.9%) and Rwanda (76.2%) followed by general household merchandise shops (39% in Burundi and 21.1% in Rwanda) (Table 4). Whereas, most of the farmers in Burundi (36.6%) asked other farmers which type of pesticide to buy, their counterparts in Rwanda depended more on their own experience (51.8%). Information about pesticide use dosages was provided mainly by the agrochemical retailers (48% in Burundi and 27.4% in Rwanda).

Routine pesticide application was more common in Rwanda (70.8%) than in Burundi (36.1%). More than a half of the respondents (62.6% in Burundi and 54% in Rwanda) reported using damaged knapsack sprayers which could increase chances of body contact with pesticides, therefore contributing to poisoning cases. Selling of pesticides in unlabeled containers was more common in Burundi (70.5%) than in Rwanda (40.8%). Only 20% of farmers in Burundi and 17.3% in Rwanda could read and understand the pesticide label. The proportion of farmers who could tell the toxicity of pesticides from its label were very low (3.4% and 13.4% in Burundi and Rwanda, respectively). Knowledge of negative impacts of pesticide use on the environment was also very low (12.6% of farmers in Burundi and 29.2% in Rwanda). Killing of domestic animals and killing of beneficial insects such as pollinator bees were the most well-known negative effects of pesticide use to the environment. These were recognized by 37.5% and 45% of the farmers in Burundi and Rwanda, respectively. Farmers also expressed the fear that exposure to pesticides may cause human diseases such as cancer.

## 4. Discussion

This study sought to provide baseline information on pest control methods and specifically pesticide use by smallholder farmers of RTB crops in Rwanda and Burundi. Here in, we highlight the toxicity of the pesticides being used, application frequency, use of PPE and knowledge about various aspects including perceived negative effects resulting from pesticide use. 

Which pesticides are used for pest and disease management in the four crops? No pesticides were applied in banana plantations because pesticide use in banana is not economical and the three most important banana diseases in the study areas (i.e., banana bacterial wilt, fusarium wilt, and BBTD) cannot be controlled with pesticides. The other banana pests and diseases such as banana nematodes and yellow and black sigatoka are not so severe at high altitude areas (above 1400 m.a.s.l in Rwanda) where this study was carried out. A similar explanation applies for not using fungicides in cassava and sweetpotato fields. Additionally, pesticides are not cheap and cannot easily be afforded by most farmers. The use of fungicides vis-a-vis insecticides followed the general trend for these two countries, with more farmers using fungicides than insecticides in potato [3]. Of all the pesticide imports in Rwanda, 75% are fungicides, mainly Mancozeb and Ridomil which are meant to be used in potato, tomato, and coffee [3]. 

Cultural control methods were quite popular across the four crops and this could be because of the relatively low cost and ease of implementation. In neighboring Uganda, a promising genetically modified (GM) potato variety (Vic 1) was tested that is resistant to potato late blight, thus having the potential to greatly reduce the amount of fungicides used in potato [7,32]. In sweetpotato, research efforts in the East African region are underway to promote the use of virus-free (clean) planting material by making clean vines available through decentralized vine multipliers [33]. In banana, together with other management methods, single diseased stem removal has also been shown to be effective in management of banana bacterial wilt on small farms in the eastern and central African countries [34]. In cassava, community phytosanitation has been recommended to be effective in managing cassava brown streak disease [35] and measures to enhance the quality of planting material are being promoted in most parts of the Great Lakes region.

### 4.1. Pesticide Active Ingredients and Toxicity Classes 

In our survey, we didn’t encounter any highly- hazardous pesticide, probably because pesticide imports now follow strict regulations and pesticides banned by the Rwandan government cannot be on the market. Previously, for Burundi the use of Aldicarb (an extremely hazardous insecticide and nematicide) and Dichlorvos (a highly hazardous insecticide and parasiticide) was reported [9]. It should be noted, however, that some farmers may have intentionally declined to show the enumerators any pesticide that has been banned for use in either country. This is because there are reports of cheap banned pesticides on the black market that are sourced from neighboring D.R. Congo [3]. Also, the fact that several pesticides were stored in unlabeled containers made it impossible for the enumerators to verify their active ingredients.

### 4.2. Chemical Pesticides Application Frequency

The average number of pesticide sprays per season observed in this study for potato (10.2 applications) is comparable to the weekly sprays that were reported for tomato in Burundi (about 12 times per season of three months) [12]. However, it should not be necessary to do weekly pesticide application as a preventive measure if pest and disease incidence and severity are below economic injury levels. Proper timing of pesticide application requires the proper pest and disease identification and constant monitoring and the failure to do this monitoring coupled with the fear of pest/disease crop loss is one key driver behind excessive pesticide application.

### 4.3. Use of Personal Protective Equipment (PPE) during Application

Use of PPE and other protective measures like bathing after spraying or observing wind direction are essential in reducing occupational risks. Lack of money to buy PPE, unavailability of PPE at the local market, plus reluctance to use them were the main reasons reported by farmers for not using PPE. Similar reasons have been reported for farmers in other African countries such as Uganda [36]. Our analysis also revealed that personal experiences from pesticide poisoning are an important factor determining the use of PPE. Furthermore, farmers with larger fields who make higher use of chemicals are more likely to own PPE, whereas farmer organizations that may offer pesticide use services seem to act as a disincentive for farmers to own and use PPE.

### 4.4. Perception and Information about Pesticide Use 

The low numbers of farmers who were aware of the negative effects of pesticide use observed in this study is probably the main reason why the majority of farmers didn’t use the recommended PPE during pesticide application. A high proportion of farmers (70.5%) are buying pesticides in unlabeled containers. Refilling and selling pesticides in unlabeled containers is a common practice of 96% of agrochemical retailers in Rwanda and Burundi [12]. One of the justifications for the continued sale of pesticides in unlabeled containers are the small quantities (about 0.5 kg for Mancozeb) in which farmers in the survey area buy pesticides. Due to the fragmented small farm sizes (<0.1 ha) coupled with low purchasing power, most small holder farmers are not able to afford large pesticide quantities at a time, like the 50 kg bags in which some chemicals are packed (for example, Mancozeb). Selling of pesticides in unlabeled containers such as beer or drinking water bottles, not only gives the opportunity for unscrupulous sellers to dispose of expired chemicals, but also can increase risk of unintended poisoning of children. The proportion of farmers (63.9%) who followed a fixed timetable to apply pesticides in RTB crops in Burundi was lower than the 89% reported among tomato and vegetable farmers by Mutshail et al. [12], probably because vegetable crops such as tomato and cabbage (*Brassica oleracea* L.) are of higher commercial value than RTB crops making the farmers less risk averse. It is normally the fear of loss that makes farmers routinely apply pesticides [36,37].

Limited information on self-reported cases of pesticide poisoning could be found in both Rwanda and Burundi. The World Health Organization reported 10 fatalities in Burundi in 2003 due to pesticide poisoning [9]. In neighboring countries, however, research showed that cases of pesticide poisoning are common [36]. Since most farmers get information about pesticides from agrochemical retailers it’s imperative to have these agrochemical retailers trained and certified to reduce cases of pesticide poisoning among farmers. Results from this study can be generalized to represent the perceptions, attitudes, and practices of all RTB farming households in Rwanda and Burundi since the sample size is sufficient to provide a good representation of the situation in the two countries [38]. 

Results of the ZINB model alluded that, although farmers in Burundi owned more PPE than their counterparts in Rwanda, they used them less. Possible explanations for this could because of the difference in the temperature. Burundi especially Bujumbura is quite hot (average maximum temperatures is 30 °C) and working in the field on sunny day with gumboots, overalls, and a hat can be very uncomfortable.

## 5. Conclusions

The research that we conducted in Rwanda and Burundi showed that while there is infrequent use of pesticides for several of the root and tuber crops (bananas, cassava, and sweetpotato), they are commonly used for potato, although more than half of pesticide users do so with no personal protective equipment. This has important deleterious consequences on the health of farmers.

We therefore recommend training of farmers and agrochemical retailers of pesticides in safe use of pesticide and handling; formation of regulations or guidelines for safe pesticide use, handling, and storage; ensuring that laws governing pesticide use are enforced, use of IPM approaches to reduce pest and disease damage, as well as the use of pictograms instead of text to teach uneducated farmers about pesticide toxicity. 

A shift from the sole reliance on pesticides towards the use of IPM approaches that are more sustainable and environmentally friendly should be prioritized and promoted through increased institutional support. Where pesticide use is essential, chemicals should be applied as part of IPM programs that recommend application only where scouting results showed that economic injury levels have been exceeded. Finally, although the use of GM remains still a controversial issue in many African countries, there are clear health and economic benefits that will be accrued from reduced pesticide use when using GM blight-resistant potatoes for potato late blight control. To realize these benefits, African countries will need to put in place appropriate biosafety policy regulations to allow farmers access to this technology. 

## Figures and Tables

**Figure 1 ijerph-16-00400-f001:**
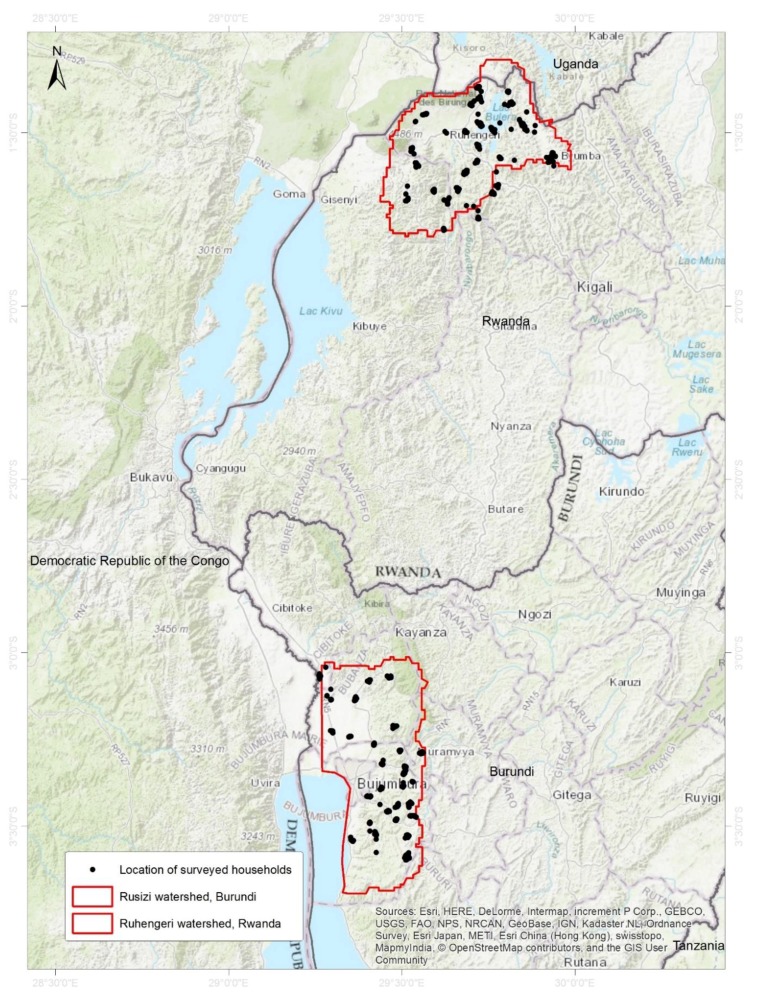
Map of Rwanda and Burundi showing the location of the watersheds and surveyed households.

**Figure 2 ijerph-16-00400-f002:**
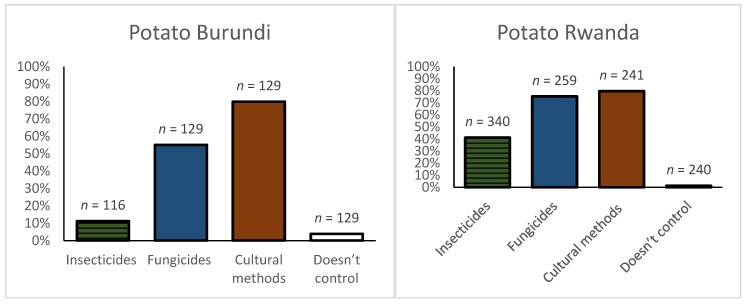
Pest and disease management practices used in potato in Rwanda and Burundi. Multiple responses were possible. *n* = number of responses.

**Figure 3 ijerph-16-00400-f003:**
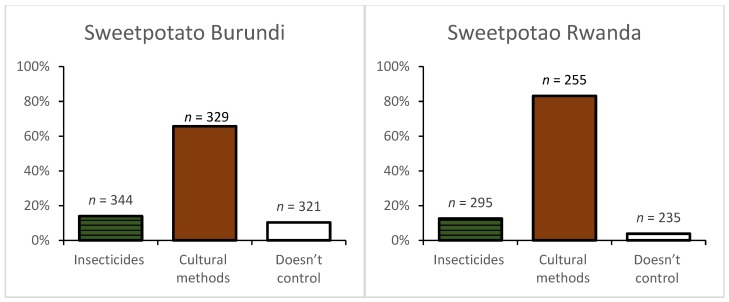
Pest and disease management practices used in sweetpotato in Rwanda and Burundi. Multiple responses were possible. *n* = number of responses.

**Figure 4 ijerph-16-00400-f004:**
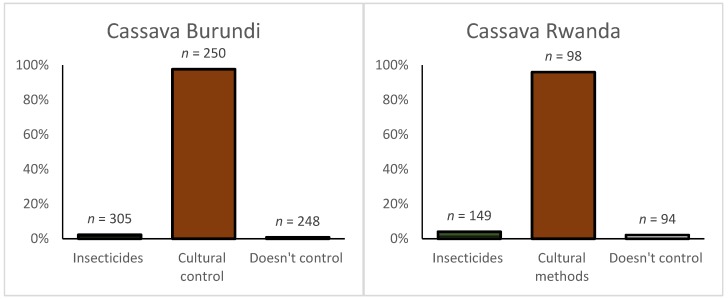
Pest and disease management practices used in cassava in Rwanda and Burundi. Multiple responses were possible. *n* = number of responses.

**Figure 5 ijerph-16-00400-f005:**
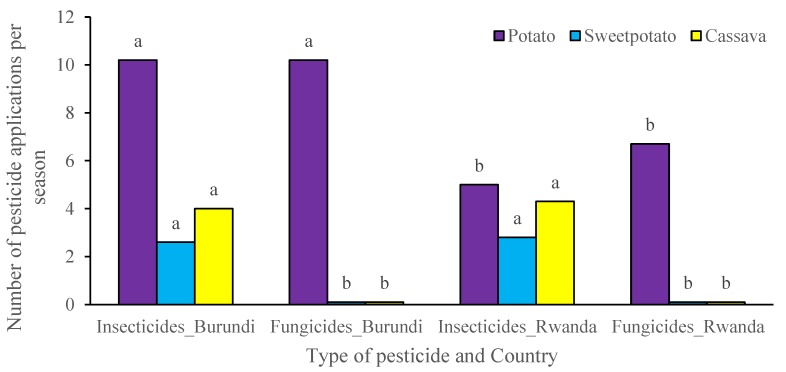
Frequency of pesticide applications by farmers of RTB crops in Rwanda and Burundi. For mean bars with the same letter per crop, no significant statistical difference at *p* ≤ 0.05 exists between countries (for the same pesticide type).

**Figure 6 ijerph-16-00400-f006:**
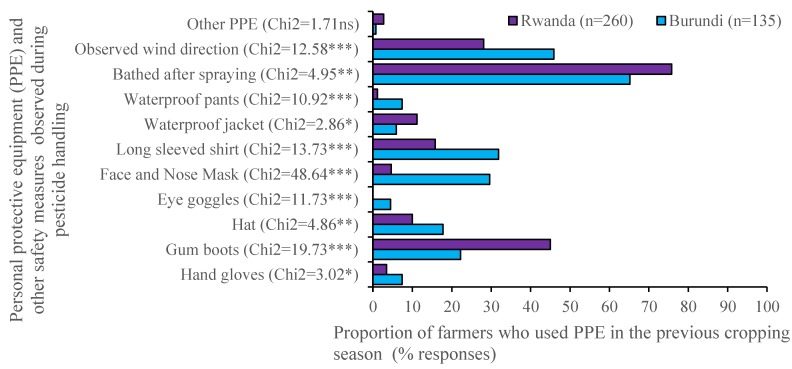
Use of personal protective equipment and other safety measures during pesticide handling by potato farmers in Rwanda and Burundi. ***, **, and * indicate statistical significance at *p* ≤ 0.01, *p* ≤ 0.05, and *p* ≤ 0.1, respectively. ns: not statistically different at *p* ≤ 0.1.

**Figure 7 ijerph-16-00400-f007:**
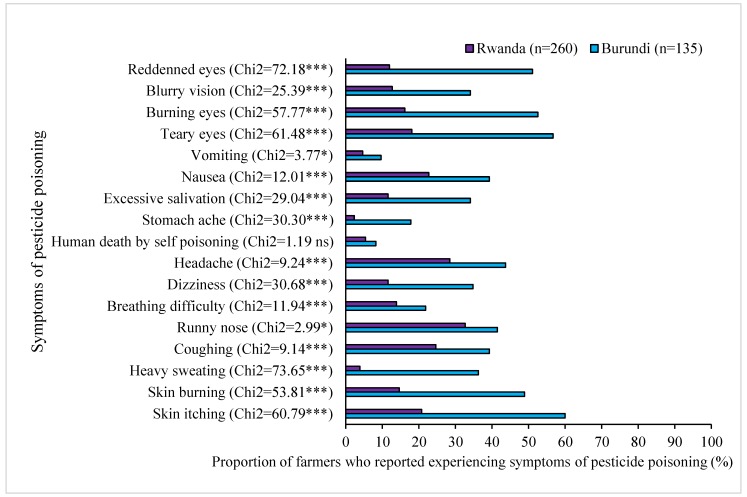
Symptoms after pesticide applications and consequences of pesticide poisoning reported by farmers in Rwanda and Burundi. ***, and * indicate statistical significance between the two countries at *p* ≤ 0.01 and *p* ≤ 0.1, respectively. ns: not statistically different at *p* ≤ 0.1).

**Table 1 ijerph-16-00400-t001:** Trade names, active ingredients, and WHO toxicity classes of pesticides used by farmers of root, tuber, and banana (RTB) crops in Rwanda and Burundi.

No.	Trade Name	Active Ingredient	WHO Toxic Class ^(a)^	Target Pest or Disease
	**Insecticides**			
1	Dursban 48 EC	Chlorpyrifos 48%	II	Sweetpotato armyworm (*Spodoptera* spp.), sweetpotato butterfly (*Acraea acerata* Hew. and the sweetpotato whitefly (*Bemisia tabaci* Gennadius) in sweetpotato Cassava mealybug (*Phenacoccus manihoti* Matile-Ferrero), cassava whitefly (*Bemisia tabaci* Gennadius) in cassava Ants (*Dorylis* spp.), aphids (*Aphis gossypii* Glover, *Aphis fabae* Scopoli, *Macrosiphum euphorbiae* Thomas, and *Myzus persicae* Sulzer), cutworm (*Agrotis* spp.), leafminer fly (*Liriomyza* spp.) and whitefly (*Bemisia tabaci*) in potato
2	Rocket 44 EC	Cypermethrin 4% + Profenofos 40%		
3	Cyper	cypermethrin 5%		
4	CyperGreen			
5	CyperLacer 5 EC			
6	Cypermethrin			
7	Dudu			
8	Dudu Cyper			
9	Dimethoate	Dimethoate 40%		
10	Tafgor 40 EC			
11	Malataf 57 EC	Malathion 57%	III	Potato tuber moth (*Phthorimaea operculella* (Zeller)) during seed potato storage
	**Fungicides**			
1	Ridomil Gold	Mancozeb 64% + Metalaxyl 4%	II	Late blight in potato
2	Emexyl	Mancozeb 64% + Metalaxyl 8%		
3	Victory 72 WP			
4	Safari max			
5	Safarizeb	Mancozeb 80%	U	
6	Dithane M 45			
7	Mancozeb 80 WP			
8	Benlate	Benomyl		

^(a)^ II: moderately hazardous; III: slightly hazardous; U: unlikely to present acute hazard in normal use.

**Table 2 ijerph-16-00400-t002:** Descriptive statistics for the variables used in the regression model.

Variables	Mean	Std.	Min	Max
Dependent variable				
Number of personal protective equipment (PPE)	0.9	1.4	0	8
Independent variables (continuous)				
Application frequency per season (APP_FRQ)	4.7	7.0	0	40
Potato field size in the current cropping season in square meters (POT_FLD)	2166.2	3063.2	0	16,200
Years of applying pesticides in potato (YRS_PST)	7.5	8.6	0	37
Years of growing potato (YRS_POT)	16.8	14.5	0	80
Total Annual Income in US $ (INC_USD)	706.8	1065.9	0	9863
Age of the head of the household (AGE_HH)	44.6	13.9	19	80
Altitude of household location in meters (ALT)	2194.4	175.3	1594	2574
Independent variables (binary)			no	yes
Application of pesticides in potato storage (APP_STORE)			287	39
Someone of the family fell sick from using pesticides (FAM_EFF)			294	32
Have you experienced any effect after pesticide applications (OWN_EFF)			159	167
Member of HH involved in a farmer organization (HH_ORG)			236	90
Received training in pest and disease management of potato (MNG_EDU)			272	54
Independent variables (categorical)				
Country of origin (CTY)	Rwanda	216		
	Burundi	110		
Formal education of the head of the HH (HH_EDU)	none	87		
	primary	185		
	secondary	54		

**Table 3 ijerph-16-00400-t003:** Summary results of the zero-inflated negative binomial model (ZINB) model and the influence of the different variables on the number of PPEs used.

Number of obs.	326	Log pseudolikelihood	−400.5391
Zero obs.	177	Wald chi-square (15 df)	111.43
Non-zero obs.	149	*p*-Value	0.0000
**Variables**	**Coef.**	**Std. Err.**	***z*-score**
APP_FRQ	0.017993	0.009113	(1.97) **
POT_FLD	0.000048	0.000022	(2.18) **
YRS_PST	0.019312	0.011964	(1.61)
YRS_POT	−0.003406	0.006742	(−0.51)
INC_USD	−0.000028	0.000062	(−0.45)
AGE_HH	0.006965	0.006059	(1.15)
ALT	0.000103	0.000470	(0.22)
APP_STORE (yes)	−0.286144	0.230425	(−1.24)
FAM_EFF (yes)	0.325450	0.333108	(0.98)
OWN_EFF (yes)	0.575926	0.180107	(3.2) ***
HH_ORG (yes)	−0.657567	0.173279	(−3.79) ***
MNG_EDU (yes)	−0.047817	0.176765	(−0.27)
CTY (Burundi)	0.737156	0.315845	(2.33) **
HH_EDU			
primary	−0.118042	0.193676	(−0.61)
secondary	0.118230	0.269461	(0.44)
Constant	−1.072239	1.104636	(−0.97)
**Variables explaining zero inflation**			
CTY			
*Burundi*	15.961540	0.665338	(23.99) ***
Constant	−15.891260	0.531466	(−29.9) ***
lnθ	−1.170119	0.405037	(−2.89) ***
θ	0.310330	0.125695	

Absolute value of z statistics in parentheses: ** significant at 5%, *** significant at 1%.

**Table 4 ijerph-16-00400-t004:** Sources of information for farmers of RTB crops regarding the use of pesticides and general awareness and pesticide use practices in Rwanda and Burundi (total number of respondents in parentheses).

Various Pesticide Parameters	% Responses		Chi^2^
	Burundi	Rwanda	
Sources of pesticides (Point-of-sale)			48.75 ***
(1) Agrochemical retailers	43.9 (123)	76.2 (223)	
(2) Agricultural extension workers	10.6 (123)	0.0 (223)	
(3) General merchandise shops	39.0 (123)	21.1 (223)	
(4) Other farmers	4.1 (123)	1.4 (223)	
(5) Weekly market	2.4 (123)	1.4 (223)	
Recommendations on type of pesticide by			16.64 ***
(1) Other farmers	36.6 (123)	29.3 (222)	
(2) Own experience	30.1 (123)	51.8 (222)	
(3) Agrochemical retailers	33.3 (123)	18.9 (222)	
Recommendations on pesticide doses			32.05 ***
(1) Not needed, can read the pesticide label	5.7 (123)	19.6 (230)	
(2) Other farmers	16.3 (123)	33.0 (230)	
(3) Not needed, own experience	30.1 (123)	20.0 (230)	
(3) Agrochemical retailers	48.0 (123)	27.4 (230)	
Pesticide use practices and general awareness about its use			
(1) Followed a fixed timetable to apply pesticides	36.1 (122)	70.8 (226)	39.40 ***
(2) Used damaged knapsack sprayers	62.6 (115)	54.0 (137)	1.89 ^ns^
(3) Pesticides purchased in labelled containers	29.5 (122)	59.2 (223)	27.81 ***
(4) Can read and understand the pesticide label	20.0 (75)	17.3 (156)	0.25 ^ns^
(5) Can tell toxicity of pesticides from its label	3.4 (117)	13.4 (217)	8.44 ***
(6) Knows the negative effects of pesticide use	12.6 (135)	29.2 (257)	13.56 ***
(7) Knowledge of alternative (non-chemical) control methods	8.2 (135)	8.2 (255)	0.53 ^ns^
(8) May cause harmful effects to humans, animals and environment			12.36 **
(8a) May cause human diseases like cancer	25.0 (16)	18.3 (60)	
(8b) May cause death of beneficial insects	25.0 (16)	45.0 (60)	
(8c) May cause death of domestic animals	37.5 (16)	11.7 (60)	
(8d) May weaken crop parts if overdosed	6.3 (16)	1.7 (60)	
(8e) May help people to commit suicide	6.3 (16)	21.7 (60)	
(8f) May pollute water sources	0.0 (16)	1.7 (60)	

*** and ** indicate statistical significance between countries at *p* ≤ 0.01 and *p* ≤ 0.05, respectively. ns: not statistically different at *p* ≤ 0.1. *n* = number of respondents.

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
