# Peer review of "Pesticide Use Practices in Root, Tuber, and Banana Crops by Smallholder Farmers in Rwanda and Burundi"

_ijerph, 2019, doi:10.3390/ijerph16030400_

Round 1
Reviewer 1 Report
This paper studied the pest and disease management practices and the type of pesticides used in four root, tuber and banana (RTB) crops in Burundi and Rwanda through in-depth interviews with a total of 811 smallholder farmers. Some important results have been obtained. However, I do have some comments and suggestions for the authors to consider:
1. First, I feel there is an absence of theory in the current version.
2. The research topic of this paper is not very clear.
3. The authors can provide some information on how they selected the respondents.
4. Are the respondents representative or not?
5. More information about the measures used in the survey (item wording, internal reliabilities where appropriate) can be provided in the paper.
6. Present a table or figure with regression coefficients on the regression model.
7. I believe the authors can do more to expand their discussion of the findings to make practical and theoretical contributions.
8. Please make sure your conclusions section discusses the original and unique contribution to knowledge, and the paper's implications and just name it as "conclusions".
Author Response
Thank you very much for taking the time to review our manuscript. We have improved the manuscript and tried our level best to adequately respond to all your comments. Please see attached file for our response. Thanks.

Reviewer 2 Report
This paper reports some interesting data on pesticide usage in Rwanda and Burundi.
The following general recommendations are made:
The introduction can be shortened and more focused on the issue of pesticide usage.
The authors refer to “pesticide poisoning” for reported symptoms which are mostly aspecific. It is recommended to be less categorical and report as “symptoms compatible with high pesticide exposure” (or similar sentence).
It is recommended to use always the name of the pesticide active ingredient(s) and not the commercial name which may eventually be indicated as the prevalent commercial formulation sold.
It seems that the most important point that can be derived from the data presented are:
- Fixed time table of pesticide application, rather than IPM
- Unlabeled pesticide formulation
These should be targeted in the discussion and have a prominent position among other issues.
Finally, it is a pity that the authors did not collect “symptoms” from banana farmers who allegedly did no use pesticides. This would have been a good control group to identify “aspecific symptoms”
Minor points:
Line 55: a reference should be provided to support the impressive 100% crop loss
Line 70: the estimation of 1 kg/hectare should be indicate if it refers to active ingredient or commercial product, and which is the source of this information
Line 71: 46.7% of what?
Line 101: DDT hasn’t been used since many year. Please find another example.
Lines 108-109: the statement that commercial sellers are “only interested in maximising profit” is authors’ opinion and accusation for which no data are provided; a more neutral sentence should be used.
Line 153: what is intended by “cultural methods”
Lines 154-155: what is intended by “healthy pests”
Lines 217-218: eight fungicides are reported but only three are named
Line 240: it should be figure 5
Line 261-261: the sentence is not clear
Lines 270-271: please explain the meaning of “self-reported cases of death”
Line 281: it should be table 2
Table 2, point 8a: pesticides are very unlikely to cause cancer in humans because they are screened before commercialization (at least in the last 40 years)
Lines 342-343: please “occupation hazards” to “occupational risks”
Author Response
Thank you for your time and effort to read through our manuscript. We have revised it and it now reads better. All your observations have been responded to in the attached file. Thanks.

Reviewer 3 Report
The authors have surveyed farmers of potato, sweet-potato and cassava of two different African countries (Rwanda and Burindi) as regards as use and customs in handling of pesticides. After assessing the information the authors concluded that training of farmers and agrochemical retailers of pesticides in the safe use, storage and handling of pesticide is highly recommendable.
The authors have sampled a high number of farmers and therefore the results of the survey seem to be highly representative of the real situation in Rwanda and Burundi. Therefore, the conclusions are well supported by the results of the survey and I have only minor suggestions for the authors before publication.
Can you please explain a little bit better section 3.2.4? I am able to understand what you mean but not to understand how you reach these conclusions.
Please, be more carefully with the use of abbreviations. Note that, even when the abbreviation was defined in the abstract, it should be defined again first time was used in the main body of the text. All abbreviations should be defined, even when they were more or less common as NGO. What is the meaning of SSA and CGIAR? It is not necessary to define an abbreviation for something cited only one time, as RAB or ISABU.
Please, use lower case for dichlorodiphenyltrichloroethane (line 101)
Minor mistake: Figure 6 stated in line 240 is indeed Figure 5
Minor mistake: Table 4 stated in line 281 is indeed Table 2
Author Response
Dear Reviewer,
We are so grateful to you for your time and effort to read through our paper. We have addressed all your comments and also improved the manuscript. Please find the attached file for our response.

Reviewer 4 Report
General Comments
This paper explores the pesticide use practices in 4 important cash and food crops for Central Africa. It aims to analyze pesticide use practices among the four crops in Rwanda and Burundi. The topic is extremely interesting, and the Authors have done an effort to explore and provide information which is not available and could be of importance for future studies. Nevertheless, as it is written now, the paper floats between an agricultural report (for an Agricultural journal), sociological study (for a humanistic journal), and Environmental/Public Health article. In order for the message and information to be transmitted in the right way the Authors need to decide in which direction they would like to take the paper, and then follow the writing practices of the area they choose.
In general, I would like to congratulate the Authors on the study they have done. A large part of the information, if presented in a better way, with a clear aim, and discussed so that the knowledge on this topic is useful also for future studies, deserves publication. Nevertheless, without a major revision of the aim, presentation of the methodology and results, discussion and conclusions, this useful information will remain impossible to understand. I also recommend a native English speaker to revise the text to increase readability.
Below you can find the major concerns and advice regarding different parts of your paper in case you would like to stay in the Public Health domain.
Introduction
The Introduction is quite lengthy and does not seem to fit well with the title and the stated aims of the paper. Several parts of the Introduction seem more fitting of a journal on agriculture, and not one on Environmental Research and Public Health. Although some of the information the Authors provide in the Introduction are useful for understanding the crops, their importance, pests, and pesticides used in the two countries, they are far too detailed and offer no synthesis or a conclusion which would be of interest for a Public health researcher. At least two times in the Introduction the Authors mention potential implications of their work, which is more suited for the Discussion and Conclusions, but it is difficult to find one clear statement regarding the aim of this paper. Several aims are mentioned in several places. Without a clear statement on the aim of the paper it is impossible to evaluate the suitability of the information provided in the Introduction, nor how the Results, Discussion, and Conclusion fulfill this aim.
Methods
A map of the areas covered and not covered by the study would really improve the part of the Material and Methods regarding the survey area. In addition, the Authors should provide, even as Supplementary material, the English version of the questionnaire which was used. Using a newly created questionnaire in studies like this one can be really challenging, as there are many studies already published on knowledge, attitudes and practices (KAP studies) using various tools, and a short review or comparison of the tools you developed with the existing literature could be helpful.
The description of the questionnaire (without being able to see the English version) is not enough to understand how the questions were asked, what were the possible answers, etc. For example, it is unclear what is the difference between “pest” and “disease” control, what is meant under “methods”, what questions are there on “toxicity” which the farmers can answer.
I leave it to the Editor to evaluate if in this case an “oral informed consent” is acceptable. I believe that if one agrees to the questionnaire/interview this automatically means they have given consent.
The way it is written now, the Statistical analysis section gives no information to the reader regarding the data analysis and creates doubt on how it was conducted. Writing that ANOVA and chi square tests were used to analyze the survey data and attaching the reference for SAS does not give sufficient information. Since the questionnaire is not provided, and the aim of the work is unclear, it is difficult to understand what will be verified by statistical tests. It seems from the tables and figures the Authors were comparing the results from Burundi and Rwanda.
Results
I sincerely advise against using “pie charts”. They are types of graphs most difficult to understand (i.e. to compare frequencies of various answers), and they take up too much space. My suggestion is to use bar charts for this.
For the purpose of Public health, the part 3.1. of the Results is too long and detailed. The information is interesting, but I suggest to the Authors to try to present the information with tables or graphs, and to underline only the most important facts – what they find important. Here I see no comparison or statistical tests to compare the differences between the two areas.
Part 3.2. is much more in the Public and Environmental Health domain. The results are very well presented in this part. I suggest you use a different mode of presenting statistically significant values in Figure 4 (not a,b). For example: https://i0.wp.com/www.sthda.com/sthda/RDoc/figure/ggpubr/add-p-values-to-ggplots-pairwise-comparisons-1.png?w=450
I cannot understand why the Authors compare Rwanda and Burundi, as from the Introduction nor from the aims it was unclear that this was one of the goals of the paper. Nevertheless, if they are compared, some explanation regarding the reason for the observed difference is necessary.
The information regarding the Zero-inflated negative binomial regression model is missing – no quantitative data is shown to accompany the text of this section. It is also unclear how the farmers (several hundred of them) report deaths of humans and animals, but this is part of the poorly described methodology for data collection. A value of this paper would be to allow for future studies to use the same methodology (questionnaire) but it is unavailable.
It is unclear to me how “flu” is considered a symptom of pesticide exposure, as it is a disease caused by the virus of flu. Also I do not agree that “death of domestic animals” can go in the same category of SYMPTOMS with stomach ache. In tables you cannot write “* indicate statistical significance between…” as this is not a correct way to report the test results.
Discussion
The discussion should be better organized and offer some comparison to available data. If data is not available, this should be clearly stated. Also it is necessary to put the data in a context to allow the reader to understand what the purpose of the study way, what were the results, and how they answer the purpose of the study.
It is unclear to me how such a large number of deaths and poisonings happens together with the use of active substances which are not rated as highly hazardous, and this should be explained or re-elaborated.
In the Discussion I miss a paragraph explaining the advantages and disadvantages of the study, limitations, and what should be studied in the future as a guideline for other authors or the Authors themselves.
In addition, it would be useful (also for the Authors) to try to summarize the main findings of their study in one paragraph. Comments on the “sufficiency” of the sample size are out of place if not enriched with the total number of farmers and their “spread” throughout the territory.
Conclusions
The Conclusions stated are just recommendations, and they could have been made even without the study. Training, safe use, regulation, etc… are general recommendations which can be given in any country, developed or developing, with more or less serious pesticide problems. The conclusions should be specific to this study, allowing us to see a clear connection between the study aims, results, discussion, and finally recommendations.
Author Response
Dear Sir/Madam,
Your review was very helpful in improving the writing of this manuscript. We have done our best to address most of your comments and we hope you will approve our manuscript to go to the next stage. Please find our response in the attached file. Thanks.

Round 2
Reviewer 1 Report
The authors have revised their manuscript based on the reviewers' comments and suggestions.
Reviewer 4 Report
Dear Authors,
I congratulate you on improving your manuscript.
I will give you just these two minor, friendly, comments and leave it to you to decide if you will follow them:
In figures, the % axes do not need the decimal place (no need to show XX.0%).
The statistical analyses section does not need to explain the mathematics behind the statistical methods you report, but just to show what statistical method was used for what analysis (e.g. Chi-square test was used for verifying associations between the crop and the protection type, etc...) and a short description if some "out of the ordinary" method was used. You provide much more information than needed for an average reader, but an advanced reader might appreciate it.